# V1Net: A computational model of cortical horizontal connections

## Abstract

The primate visual system builds robust, multi-purpose representations of the external world in order to support several diverse downstream cortical processes. Such representations are required to be invariant to the sensory inconsistencies caused by dynamically varying lighting, local texture distortion, etc. A key architectural feature combating such environmental irregularities is long-range horizontal connections that aid the perception of the global form of objects. In this work, we explore the introduction of such horizontal connections into standard deep convolutional networks; we present V1Net – a novel convolutional-recurrent unit that models linear and nonlinear horizontal inhibitory and excitatory connections inspired by primate visual cortical connectivity. We introduce the Texturized Challenge – a new benchmark to evaluate object recognition performance under perceptual noise – which we use to evaluate V1Net against an array of carefully selected control models with/without recurrent processing. Additionally, we present results from an ablation study of V1Net demonstrating the utility of diverse neurally inspired horizontal connections for state-of-the-art AI systems on the task of object boundary detection from natural images. We also present the emergence of several biologically plausible horizontal connectivity patterns, namely center-on surround-off, association fields and border-ownership connectivity patterns in a V1Net model trained to perform boundary detection on natural images from the Berkeley Segmentation Dataset 500 (BSDS500). Our findings suggest an increased representational similarity between V1Net and biological visual systems, and highlight the importance of neurally inspired recurrent contextual processing principles for learning visual representations that are robust to perceptual noise and furthering the state-of-the-art in computer vision.

## 1 Introduction

Following Hubel and Wiesel's (Hubel & Wiesel, 1968) seminal work on characterizing receptive fields in the cat striate cortex and Fukushima's Neocognitron (Fukushima, 1980) (a hierarchical extension of this building block), two broad families of visual models have been developed by the neuroscience and computer vision communities respectively. The former family of models aims to account for findings from single-cell neurophysiology, either by directly modeling types of neuronal responses (De Valois et al., 1982; Sullivan & De Sa, 2006; Chichilnisky, 2001; Pillow, 2007) or by proposing computational models that give rise to similar neural phenomena (Olshausen & Field, 1997; Schwartz et al., 2006).

The latter family of models, particularly Deep Convolutional Networks (DCNs) (LeCun et al., 1998), are also loosely inspired by Hubel & Wiesel (1968) and Fukushima (1980); they aim to optimize performance on a wide range of computer vision benchmarks including but not limited to image recognition (Krizhevsky et al., 2012; Hu et al., 2018; Simonyan & Zisserman, 2014), contour detection (Xie & Tu, 2015; Shen et al., 2015) and object segmentation (He et al., 2017; Long et al., 2015). Despite their impressive performance on benchmarks in these areas, these models progressively deviate from biological vision and recent work highlights their interesting deficiencies and sensitivities relative to primate vision (Geirhos et al., 2019; Papernot et al., 2016; Kurakin et al., 2016; Eykholt et al., 2017).

Our motivation is to reverse-engineer the cortical contextual processing principles that have been shown to be mediated by long-range horizontal connections using DCNs. Horizontal connections are capable of systematically filling-in sensory inconsistencies and spatially binding neighbouring features together in order to provide stable perception. In an attempt to complement DCNs with this property and to contribute towards bridging the gap between artificial and biological visual representations, we develop 'V1Net', a novel recurrent unit inspired by visual neuroscience and Gestalt psychology literature on cortical horizontal connectivity (Das & Gilbert, 1995; Grossberg & Mingolla, 1985). V1Net can be flexibly incorporated as a module in existing implementations of DCNs.

In the following Section 2, we briefly survey the existing literature on biologically plausible vision models that are related to V1Net. In Section 3, we introduce the V1Net model along with it's mathematical formulation and an intuitive explanation of its working. In Section 4, we demonstrate experimental results from: (1) Texturized Challenge, our proposed benchmark to evaluate object recognition ability under perceptual noise and (2) an ablation study of V1Net's horizontal connections using the BSDS500 boundary detection benchmark (Arbelaez et al., 2011). Subsequently, we demonstrate several biologically plausible horizontal connections emerging in a V1Net while learning the task of boundary detection. We also share qualitative results of V1Net's zero-shot domain transfer on the task of object boundary detection from natural images (used for training) to stylized images (Gatys et al., 2015). Finally, we discuss our current plans for extending the work.

## 2 RELATED WORK

Computational models of visual perception have been an interesting and active area of research in the vision science and machine vision communities. Riesenhuber & Poggio (1999) and Serre et al. (2007b;a) introduced hierarchical visual models inspired by computations in the visual cortex to perform robust object recognition. Relevant to their models is the recently proposed CORNet by Kubilius et al. (2018). CORNet is a deep recurrent-convolutional model of the ventral visual stream primarily motivated by the task of core object recognition. There has been a parallel direction of research related to cortically inspired models, such as Fan et al. (2018) that compare the representational similarity between the ventral visual stream and hierarchical deep convolutional models trained on image recognition. Concurrently, there have been several efforts such as Cao et al. (2018); Wang et al. (2018) from the computer vision community on incorporating long-range spatial dependencies and lateral inhibition into deep convolutional networks.

There has been a great deal of interest in exploiting the potential of dynamical systems to model early visual perception which is highly relevant to our work. Early computational models of visual routines proposed by Grossberg & Mingolla (1985); Grossberg & Williamson (2001); Li et al. (2006) were primarily implemented in this dynamical systems framework using partial differential equations. However with the recent hardware and software improvements (Jouppi et al., 2017; Abadi et al., 2016) to efficiently train Recurrent Neural Networks (RNNs), there has been a surge in the development of biologically plausible models of vision using RNNs from the neuroscience and computer vision communities. Kar et al. (2019) provide experimental and computational modeling results demonstrating the need for recurrence in core object recognition routines. Through a large-scale neural architecture search over models that vary along the dimensions of their core operations, Nayebi et al. (2018) demonstrate the emergence of architectures equipped with recurrent and feedback interactions to be ideal for object recognition. Additionally, they provide evidence for the model's neural plausibility by explaining neural data recorded from the visual cortex of awake primates during image-viewing.

The horizontal Gated Recurrent Unit (hGRU) introduced by Linsley et al. (2018) is relevant to our proposed V1Net. However, there are several key differences between hGRU and the V1Net in terms of their design and horizontal connection dynamics. Among other differences, V1Net's computations are not deliberately divided into two sequential within-timestep stages. In hGRU the inhibitory horizontal connections are developed in the first stage which are followed by excitatory horizontal connections. However in V1Net, linear excitatory and inhibitory connections are not exercised in any particular order. As a subtle form of ordering, our nonlinear shunting inhibition mechanism is implemented disynaptically as they inhibit the excitatory cells which synapse onto simple cells in

a facilitatory fashion (see Eqn.1). We find this design to be more in line with biological horizontal connections.

## 3 V1NET: A MODEL OF LONG-RANGE HORIZONTAL CONNECTIONS

We employ recurrent-convolutional neural networks to model V1Net's computation. V1Net consists of a population of neurons akin to simple cells in V1 that learn to detect features from their presynaptic input (available in the convolutional activation maps from the corresponding bottom layer). Each 'simple-cell' receives recurrent inhibitory and excitatory influences from its neighboring cells which we implement using the following three types of horizontal connections: (1) Linear additive excitation, (2) Linear subtractive inhibition, and (3) Nonlinear divisive shunting inhibition. These three types of horizontal connections create a push-pull integration at each neural site, resulting in a net inhibition/excitation of a neuron's activity as a nonlinear function of its neighbors' activity.

### 3.1 MATHEMATICAL FORMULATION OF V1NET

We derive the V1Net cell by starting from the convolutional variant of a Long Short Term Memory (LSTM) cell (Hochreiter & Schmidhuber, 1997) called the ConvLSTM (Xingjian et al., 2015). The hidden-state dynamics of a ConvLSTM are governed by that of the original LSTM update equations with the fully-connected operations replaced with convolution operations. At any discrete timestep $t$ of recurrent processing, we utilize the previous convolutional layer's activation map to set the value of our recurrent input tensor at time $t$, $X_t$; i.e, the input tensor $X_t$ is set to the bottom layer's feature representation at all timesteps $t = 0$ to the final timestep $t = T$.

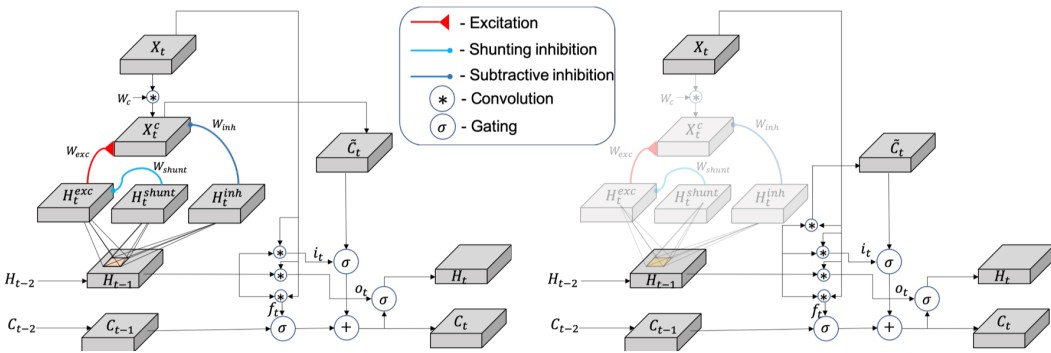

Figure 1: Comparison of a V1Net cell (left) with a ConvLSTM cell (right)

V1Net's different horizontal connections are implemented using distinct convolutional filter banks with varying spatial dimensions, namely $W_{shunt}$, $W_{inh}$ and $W_{exc}$. These convolutional filter banks model the horizontal connectivity between all possible pairs of feedforward 'simple-cells' (or convolution feature maps) giving us the ability to visualize the nature (inhibitory/excitatory) of the horizontal connections along with their spatial profile and connectivity strength between any pair of feedforward receptive fields.

Horizontal connection kernels convolve over the hidden state of V1Net, $H_{t-1}$ to produce $H_t^{shunt}$, $H_t^{inh}$ and $H_t^{exc}$ respectively. The activity at each position $H_t^{exc}[i, j, k]$ in layer $L$ denotes the net horizontal excitation received by a feedforward neuron at spatial location $(i, j)$ in the $k^{th}$ activation map from its spatial neighbors within a fixed distance from all activation maps in layer $L$. Similarly, $H_t^{inh}$ and $H_t^{shunt}$ encode the pointwise net contextual linear and nonlinear inhibition received by the feedforward neurons. Inhibitory and excitatory influences are then integrated with the input at the current timestep to produce $\tilde{c}_t$, the candidate memory cell at time $t$. $\tilde{c}_t$ is mixed with V1Net's memory cell from the previous timestep to compute $c_t$, the current memory cell of V1Net. The current hidden state $H_t$ of V1Net is computed as a gated nonlinear function of the cell state $c_t$. A key property to note is that receptive fields of neurons in the horizontal kernels grow at every iteration of recurrence, and hence their range of horizontal connectivity scales proportionate to the amount of recurrent processing. The internal working of our proposed V1Net unit is shown in Fig.1 (left) alongside that of a standard ConvLSTM (right).

The following equations summarize the working of V1Net:

$$
\begin{aligned}
f_t &= \sigma(W_f * X_t + U_f * H_{t-1}) \\
i_t &= \sigma(W_i * X_t + U_i * H_{t-1}) \\
o_t &= \sigma(W_o * X_t + U_o * H_{t-1}) \\
H_t^{exc}, H_t^{inh} &= \sigma(W_{exc} * H_{t-1}), \sigma(W_{inh} * H_{t-1}) \\
H_t^{shunt} &= \sigma(W_{shunt} * H_{t-1}) \\
\tilde{c}_t &= \boldsymbol{\alpha} \times H_t^{shunt} \times (W_c * X_t + H_t^{exc}) - \boldsymbol{\beta} \times H_t^{inh} \\
c_t &= f_t \odot c_{t-1} + i_t \odot \sigma(\tilde{c}_t) \\
H_t &= o_t \odot tanh(c_t)
\end{aligned}
\tag{1}
$$

Each $W_.$ and $U_.$ is a 2-D convolution kernel, '$*$' and $\sigma(.)$ represent 2D convolution operation and the sigmoid nonlinearity respectively. $\boldsymbol{\alpha}$ and $\boldsymbol{\beta}$ are learnable parameter vectors that allow V1Net to selectively scale up/down horizontal influences in a channel-wise fashion. Additionally as a straightforward mathematical way to compare a V1Net cell to a ConvLSTM cell, we present below the state-update equations of a standard ConvLSTM cell.

$$
\begin{aligned}
f_t &= \sigma(W_f * X_t + U_f * H_{t-1}) \\
i_t &= \sigma(W_i * X_t + U_i * H_{t-1}) \\
o_t &= \sigma(W_o * X_t + U_o * H_{t-1}) \\
\tilde{c}_t &= \sigma(W_c * X_t + U_c * H_{t-1} + b_c) \\
c_t &= f_t \odot c_{t-1} + i_t \odot \sigma(\tilde{c}_t) \\
H_t &= o_t \odot tanh(c_t)
\end{aligned}
\tag{2}
$$

From (1) and (2), it can be observed that the equations governing the representation dynamics of V1Net and ConvLSTM are fairly similar. We also present diagrammatically, the process of converting a ConvLSTM into a V1Net by adding the transparent components in Fig.1 (right).

## 3.2 PICTORIAL EXAMPLE DEMONSTRATING V1NET'S WORKING

By design in DCNs, the flow of information between neurons within a particular spatial resolution (within a layer) is restricted to the dimensions of the convolution kernels in that layer. This may not be ideal for tasks such as object recognition in the presence of perceptual noise (such as clutter) or contour tracing wherein long-range neighbouring elements at a particular spatial resolution should explicitly be grouped together/separated in order to form a holistic object representation. V1Net's long-range horizontal connections precisely build in this functionality of grouping within a DCN layer in order to efficiently link elements and represent objects robustly.

Consider the toy example of multi-digit recognition under perceptual noise (breaking up of object boundary) and clutter as shown in Fig. 2. Standard DCNs that learn this task will require explicit supervision for every possible configuration of overlapping digits in order to classify both digits correctly. Moreover, one may suspect that a DCN may not generalize during test-time to overlapping digit pairs that were not presented together during training time. This is because of the lack of a DCN's ability to group together elements that belong to an object and separate out those that belong to different objects.

In a V1Net model, the horizontal excitatory connections (highlighted in red in Fig. 2) learn to selectively link elements that are like-oriented in a neighborhood, resulting in a strong facilitation and representational grouping of those elements that belong to a digit. On the other hand, inhibitory connections (highlighted in blue) learn to suppress elements that are not like-oriented and (probably) don't belong to the same object, hence perceptually decluttering and separating the different target digits to be recognized. Due to this reason, a V1Net model will require many fewer examples to learn to recognize digits under clutter as it only needs to learn invariant digit templates on top of the horizontal connections, and not require rote-memorization of every possible configuration of overlapping digits. For the same reasons a V1Net model will require fewer parameters than a standard DCN at a particular performance level, as the DCN needs a large number of parameters to memorize all object configurations. Due to this decluttering effect of V1Net, we hypothesize that it will also generalize to unseen pairs of overlapping/cluttered objects during test-time.

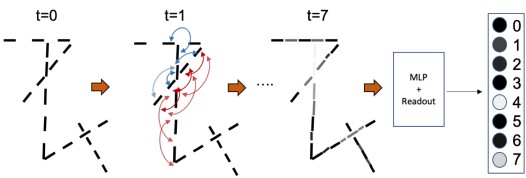

Figure 2: Toy example showing excitatory grouping (in red) and inhibitory decluttering (in blue) during digit recognition under clutter. Output represents the predicted probability distribution over all digits (light shade represents high probability). Transparency of horizontal connections reflect their strength.

## 4 EXPERIMENTS

We evaluate the proposed V1Net model on two different tasks related to perceptual robustness and early visual representations. First, we describe the Texturized challenge, a novel challenge we present along with this paper that aims to assess a model's ability to robustly recognize objects under an identity-preserving image transformation based on Fourier analysis. Second, we show experimental results from a V1Net model trained to perform the task of boundary detection on natural images from the Berkeley Segmentation Dataset, BSDS500. Please refer to Appendix A for details about the architectural design for these experiments.

### 4.1 TEXTURIZED CHALLENGE

Given the ability of horizontal connections to provide robustness against local perceptual noise, we designed a challenge that we call the 'Texturized challenge'. This challenge aims to assess the perceptual robustness of various models by testing their ability to recognize object shape under perceptual noise and uncertainty in their texture. All images used to generate the stimuli for texturized challenge were sourced from the CIFAR 10 object recognition dataset (Krizhevsky et al., 2009).

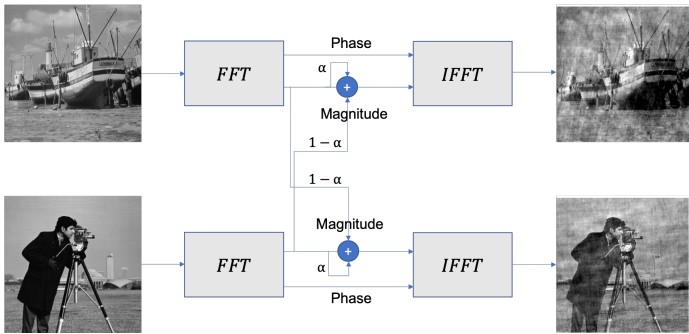

Figure 3: A demonstration of our process used to generate Texturized Challenge images

By applying a Fourier transform on RGB images, we obtain a representation of the image in Fourier space separating its phase and magnitude signals. One may incorporate local perceptual noise into an image, particularly in the object texture (Morgan et al., 1991) while retaining the overall object identity (via object form) by reconstructing the image with an intact phase and perturbed magnitude representation. However, corrupting the phase (which is of higher importance for object identity (Oppenheim & Lim, 1981; Piotrowski & Campbell, 1982)) and retaining the magnitude results in a reconstructed image with disrupted object identity, recognizing objects from such images is difficult even for humans and is hence not considered for our challenge. By exploiting this conceptual framework, we generate images for our Texturized Challenge. Texturized images for each sample from CIFAR10 are obtained by reconstructing the sample using its own Fourier phase representation, and a linear combination of its own Fourier magnitude with the Fourier magnitude of a different randomly selected image from CIFAR10. This selectively adds local perceptual noise to objects present in the CIFAR10 dataset and retains the overall shape of objects. This complete procedure

is demonstrated in Fig. 3. As observable from the inset figures in Fig. 4, humans are fairly robust to this challenge. The ratio that governs the amount of randomly selected Fourier magnitude denotes the difficulty levels in the challenge; we present results for 4 increasing difficulty levels in our evaluation below.

For this challenge, we systematically compare V1Net to 2 other parameter matched control models that are void of horizontal connections. Each model was developed using the above described 3 layer framework, with the third layer being one of: (1) V1Net with horizontal connections, (2) A traditional ConvLSTM model without horizontal connections (ConvLSTM), and (3) A feedforward baseline with a regular 2D convolution without horizontal connections (Control-FF).

We evaluated the test-time classification accuracy of the above three types of models on images from the Texturized challenge, the results for this evaluation are presented in Fig. 4. Each model was tested on a range of 4 different versions of the task with increasing difficulty levels plotted on the vertical axis. The horizontal axis represents the test-set accuracy of models at different levels of task difficulty, which we measure using the amount of noise present in the Fourier magnitude.

From the below evaluation, we observe two attributes related to the efficacy of recurrence and horizontal connections: (1) Feedforward models are the least robust to perceptual noise compared to recurrent models. This is validated by the feedforward control's quick deterioration in classification accuracy at a relatively easier version of the task (compared to recurrent models). This result provides positive evidence to the claim that recurrent processing aids visual recognition, and is capable of adding robustness to visual representations. (2) The results also reflect the superior performance of the V1Net model with horizontal connections compared to the control models. V1Net's improved robustness relative to also the recurrent control (ConvLSTM) highlights the utility of horizontal connections in enhancing the robustness of a model's object representations.

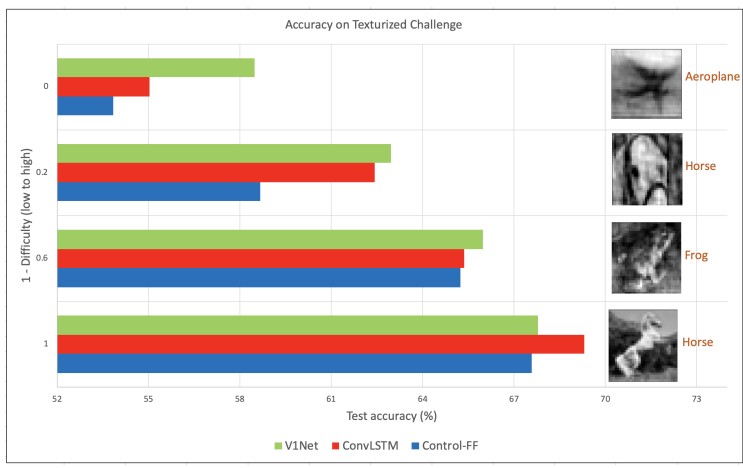

Figure 4: Performance of (1) V1Net, (2) Recurrent control (ConvLSTM) and (3) Feedforward control models (Control-FF) on the Texturized challenge. Note that the difficulty level under the green bar is the same difficulty for the adjacent blue and red bars.

## 4.2 UTILITY OF HORIZONTAL CONNECTIONS TO DETECT OBJECT BOUNDARIES – ABLATION STUDY

We evaluate the proposed V1Net model on the task of boundary detection on natural images from the Berkeley Segmentation Dataset, BSDS500 (Arbelaez et al., 2011). We chose a boundary detection task due to the correspondence of our model with early layers in the primate visual cortex that detect contours from visual input among their other functions. We compare 3 different lesions of V1Net for this ablation study: (1) A standard ConvLSTM without horizontal connections acts as our comparison baseline with all horizontal connections lesioned. We test 2 variants of this architecture with (a) wider convolution kernels (ConvLSTM-LargeRF) and (b) greater number of channels (ConvLSTM-MedRF) to match effective receptive field and/or number of parameters with

V1Net respectively. (2) We evaluate the necessity of nonlinear inhibition in the presence of linear horizontal connections by testing a lesion of V1Net without shunting inhibition (V1Net-Linear).

Fig.9 shows a sample set of boundary predictions (from V1Net) on images from the BSDS500 test set. From our experiments with V1Net, we have consistently noticed that V1Net has relatively higher precision compared to parameter-matched control feedforward and recurrent models trained in similar environments.

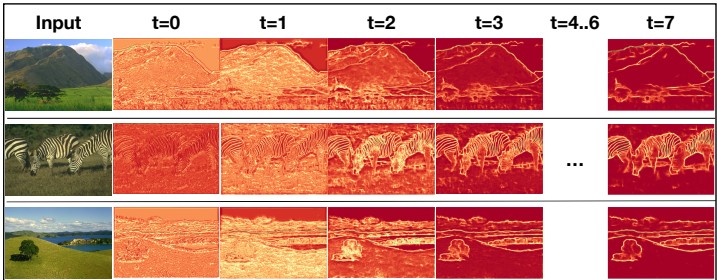

Figure 5: Evolution of V1Net boundary predictions through time on BSDS500 test set

| Model | ODS-F | OIS-F | AP | nParams |
|---|---|---|---|---|
| Human | 0.803 | 0.803 | - | - |
| ConvLSTM-LargeRF | 0.690 | 0.713 | 0.615 | 1845921 |
| ConvLSTM-MedRF | 0.653 | 0.671 | 0.581 | 527009 |
| V1Net-Linear (t=2) | 0.604 | 0.619 | 0.526 | **470241** |
| V1Net-Linear (t=4) | 0.669 | 0.687 | 0.633 | 470241 |
| V1Net-Linear (t=6) | 0.681 | 0.699 | 0.628 | 470241 |
| V1Net (t=6) | 0.710 | 0.727 | 0.659 | 520513 |
| **V1Net (t=6)\*** | 0.738 | 0.753 | 0.711 | 520513 |
| CEDN (Yang et al., 2016) | 0.788 | 0.804 | - | 119.6M |
| BDCN (He et al., 2019) | **0.806** | **0.844** | **0.890** | 16.3M |
| HED (Xie & Tu, 2015) | 0.788 | 0.808 | 0.840 | 14.7M |

Table 1: Quantitative evaluation of BSDS; * - V1Net model trained with data augmentation and larger batch size

As suggested by the Optimal Dataset Scale F-1 scores (ODS-F1) in Table.1, horizontal connections in general are helpful to increase the precision and accuracy of boundary detection. While we have not included quantitative results for a parameter-matched feedforward baseline, we noticed its performance to be qualitatively worse than ConvLSTM-MedRF, our parameter-matched recurrent baseline. These two observations together suggest the importance of recurrent processing for detecting object boundaries, and further the advantage of employing horizontal connections to push forward the prediction performance. Comparing our results from a full V1Net and one with horizontal nonlinear connections lesioned (V1Net-Linear) reveals the pivotal role played by nonlinear horizontal connections in the process of boundary detection. In addition, progressively decreasing boundary prediction performance of two variants of V1Net-Linear with shorter amounts of recurrent processing (t=4 & t=2, 4 & 2 timesteps respectively) reveals a speed-accuracy tradeoff that is reminiscent of similar previously reported observations in primate behavioral studies.

In order to assess the impact of standard techniques on improving performance on computer vision benchmarks, we trained a V1Net model (t=6) on the BSDS500 training set with online data augmentation and a $1.5\times$ larger batch size. This experiment resulted in improving V1Net's performance on all three evaluation metrics, bringing the performance closer to that of state-of-the-art object boundary detection methods (implementation details w/ hyperparameter choices can be found in Appendix. A).

### 4.3 EMERGENT NEURALLY PLAUSIBLE HORIZONTAL CONNECTIONS

There have been several accounts of 'center-on surround-off' surround modulation patterns in the visual cortex of primates perceiving natural scenes. We visualized the learned horizontal connection

patterns in a V1Net model trained to perform boundary detection on natural images. The connection patterns reveal emergent 'center-surround' and 'association-field' like receptive fields, validating V1Net's increased biological plausibility. In addition, we also observed horizontal connections giving rise to border-sensitive cells in our model that we hypothesize to correspond with 'border-ownership' cells (Hesse & Tsao, 2016). Such connections in our model horizontally facilitate activity of cells with receptive fields (RFs) on a particular side of an object boundary (inside/outside the object) and inhibit those with RFs on the opposite side of the boundary. We believe such cells are enabling the model to confidently predict boundaries of objects by providing strong signals of the object's extent and suppress those cells in their surround that represent spurious boundaries. A sample of the biologically realistic horizontal connections learned by V1Net while performing boundary detection on natural images is presented in Fig. 6.

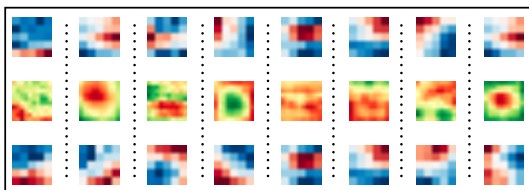

Figure 6: Horizontal connectivity patterns learned by V1Net. Top and bottom rows show learned convolution kernels in V1Net (simple-cells). The middle row shows spatial layout of learned horizontal connectivity patterns between the top and bottom cells (synaptic strength increases from blue/green to red).

### 4.4 ZERO-SHOT GENERALIZATION TO STYLE-TRANSFERRED STIMULI

In a separate experiment, we tested how well our V1Net model trained on detecting boundaries from natural images generalizes to style-transferred stimuli (top row in Fig. 7) from Stylized-Imagenet (Geirhos et al., 2019). Results from this experiment suggested that indeed, V1Net was capable of zero-shot generalizing to robustly detect object boundaries from Stylized-Imagenet. Since there isn't a way to quantitatively analyze this experiment (lack of ground-truth data), we have included a sample of V1Net's predictions on Stylized-Imagenet below. We hypothesize that V1Net's long-range horizontal connections enable it to learn global features that are spatially distant. Hence, V1Net does not fail to detect boundaries, without explicit supervision, despite the texture of objects being stylized.

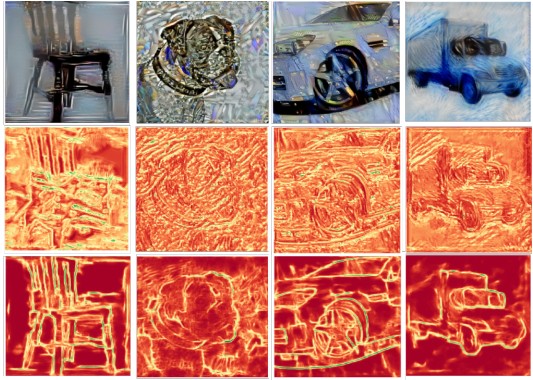

Figure 7: Zero-shot generalization to Stylized-Imagenet boundary prediction. Middle row: Boundary predictions without recurrent processing showing edges induced by fake-texture (low precision); Bottom row: Predictions from V1Net post recurrent processing emphasizing the true object edges over the added-texture edges (high precision).

## 5 DISCUSSION

Horizontal connections have been a key interest to the field of visual neuroscience, owing to their diverse functionality that gives rise to invariant visual representations. To this end, we develop a neurally-inspired recurrent neural network model of long-range horizontal connections and demonstrate its importance also to artificial vision systems for giving rise to robust visual representations through a systematic comparison against parameter-matched control models on our proposed Texturized challenge and the BSDS500 object boundary detection benchmark. We subsequently demonstrate the emergence of biologically plausible horizontal connection patterns from V1Net suggesting a strong representational similarity between V1Net and primate early visual areas. We also present qualitative results of a V1Net model generalizing in a zero-shot fashion to images with stylized texture. Continuing in this direction of using inspiration from biological vision to advance artificial vision, we are currently working on incorporating learned top-down feedback connections in a deep convolutional network along with V1Net. We believe that such re-entrant connections along with horizontal connections will allow smooth flow of information within- and across different spatial resolutions of the feature hierarchy contributing to more robust and efficient semantic segmentation models that generalize better than their feedforward counterparts. In parallel, we are actively working on quantitatively analyzing the match between V1Net's internal representations and single-cell experimental recordings collected from the primate visual cortex.

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

## A  IMPLEMENTATION DETAILS

### A.1  ABLATION STUDY ON BSDS500

The architecture we use to develop models for our experiments fits into a 3 layer hierarchical architecture that we parallel with the early visual circuit from Retina to V1. While we refer to the different layers in our model with early visual areas, the retina and LGN in our experiments are not designed based on the working of their biological counterparts.

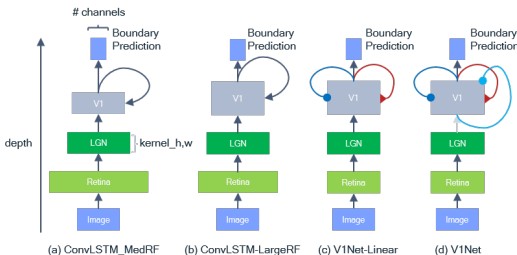

Figure 8: Architectural details from the ablation study on BSDS500

All the above four models have 3×3 2D-convolution layers in their Retina and 5×5 2-D convolution layers in their LGN. Each model uses a different number of filters in their retina and LGN ranging from 32 to 96 depending on the number of parameters in their V1 layer.

Our ConvLSTM models are implemented using TensorFlow's built in ConvLSTM2D layer, ConvLSTM-MedRF (Fig.8.a) utilizes 8×8 horizontal connection kernels, while ConvLSTM-LargeRF (Fig.8.b) utilizes 15×15 convolution kernels. V1Net-Linear (Fig.8.c) and V1Net use 15×15 2-D convolutions for their horizontal excitation and 7×7 2-D convolutions for their horizontal inhibition. The spatial dimensions are chosen to be larger for excitatory connections than for inhibitory connections to facilitate short-range inhibitory and long-range excitatory connections. In addition, the complete V1Net model (Fig.8.d) employs another set of 5×5 2-D convolutions for the shunting horizontal connections.

Similar to Zamir et al. (2017), we set the recurrent input $X_t$ to the feedforward convolutional feature representation from the bottom layer. For eg., when a V1Net block is plugged between conv-2 and conv-3, the activation map of conv-2 is used to set $X_t, \forall\, t \in [0, T-1]$.

We use the TensorFlow framework to implement all our models in this paper. We found t=6 timesteps to be suitable for our V1Net and ConvLSTM models to produce accurate object boundaries while also fitting in our device's memory. We are currently evaluating whether a larger number of timesteps than 6 is useful to improve performance further using higher end training resources. Our hyperparameter choices include: minibatch size (6), learning rate (1e-4), optimization algorithm (rmsprop), learning rate decay (divide by 10 after every 1000 iterations). For the BSDS500

experiments with data augmentation (last row in Table 1, we used 5 different data augmentation techniques to expand the BSDS training set. We utilized random brightness variation, random contrast variation, random rotation between $-\pi/4$ to $\pi/4$, horizontal flipping and vertical flipping. All these augmentation techniques were also applied to our Texturized challenge experiments. Our models are trained on a single NVIDIA 2080 GPU.

## A.2 TEXTURIZED CHALLENGE ARCHITECTURE DETAILS

The Retina in all three models had 2 3×3 convolution layers, and the LGN utilized 2 3×3 convolutions followed by a max pooling layer of size $2 \times 2$ with stride 2. Since CIFAR-10 images are much smaller in size than BSDS500 (10x smaller), we reduced the size of our convolution kernels in the last layer, V1Net to 6×6 for $W_{exc}$ and 4×4 for $W_{inh}$ and $W_{shunt}$ respectively. For our control ConvLSTM model, we set the size of the convolution kernels in the last layer to 6×6.

## A.3 MORE SAMPLE PREDICTIONS OF V1NET ON BSDS500 TEST SET

Attached below is a set of boundary predictions from V1Net on the BSDS500 test set.

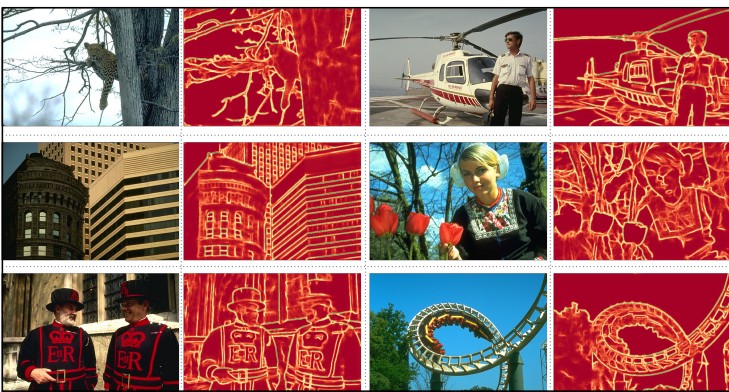

Figure 9: Sample boundary predictions from a V1Net model from the BSDS500 test set

