# OpenReview forum: "V1Net: A computational model of cortical horizontal connections"
_ICLR.cc/2020/Conference — Reject_

### Official Review · AnonReviewer2 · 2019-10-23
**Official Blind Review #2**

**Rating:** 1

**Review:**

The authors propose to modify a convolutional variant of LSTM (ConvLSTM) to include horizontal connections inspired by known interactions in visual cortex: excitation, subtractive inhibition (linear) and shunting multiplicative gating (nonlinear). They evaluate their V1Net model on a new task they call the texturized challenge (spectrally perturbed CIFAR-10 images) and on contour segmentation on BSDS500 and show that their approach outperforms some baselines.

Strengths:
+ The biological motivation is quite clear
+ Architecture is simpler than that of previous related work (hGRU)

Weaknesses:
- Not clear what the objectives/contributions are
- No advancement of state of the art in computer vision
- No novel insights about brain function
- Motivation of the "texturized challenge" is unclear
- Performance on BSDS500 is far from state of the art
- Value of the qualitative analysis on stylized ImageNet is unclear

Overall, I'm not sure what the goal of this paper is. It neither presents an advance of the state of the art in any computer vision problem nor does it lead to any novel insights about the brain. The lack of a clear statement about the contributions of the paper seems to confirm this impression – the authors don't seem to know either.

**Experience Assessment:**

I have published in this field for several years.

**Review Assessment: Checking Correctness Of Derivations And Theory:**

I assessed the sensibility of the derivations and theory.

**Review Assessment: Checking Correctness Of Experiments:**

I assessed the sensibility of the experiments.

**Review Assessment: Thoroughness In Paper Reading:**

I read the paper at least twice and used my best judgement in assessing the paper.

---

> ### Author Response · Authors · 2019-11-15
> **Official authors response to R2**
>
> We thank you for taking the time in carefully reviewing our submission. We shall address your concerns with regard to our work below:
>
> (1) Advancement of SOTA in computer vision: We acknowledge this valid concern; upon adding data augmentation and training on larger batch sizes, we were able to improve V1Net's performance as shown in the last row of Table 1. However, as shown in our Table. 1, V1Net with ~500k parameters performs closely to several SOTA boundary detection methods with orders of magnitude more number of parameters. We also do not utilize ImageNet pretraining while training V1Net on BSDS500, we expect such common practices used by the computer vision community to boost our performance; we will include results from our experiments in this direction in a future revision.
>
> (2) Motivation for texturized challenge: As mentioned in our response to R1, we proposed to use FFT texturization as one way to evaluate robustness as we find evidence in previous vision science literature on how phase information in images is crucial for humans to perceive natural images [1,2] relative to amplitude/magnitude information. We do not intend to claim that FFT texturization is the objectively correct way to evaluate perceptual robustness, however, our choice of such a manipulation is explained by previous use of this manipulation in vision science literature and the robustness of humans to this manipulation.
>
> (3) Value of OOD generalization to stylized stimuli: In this result, we wanted to demonstrate how recurrent horizontal connections improve the ability of feedforward deep nets to generalize/domain transfer beyond the training data distribution without explicit supervision. We found it interesting that a V1Net that was not trained to predict boundaries on style-transferred stimuli could invariantly regardless predict object boundaries with a reasonable precision. We are currently working on performing this experiment with quantitative results (and comparisons with SOTA boundary detection models); we hope that these results will add more value to our OOD generalization experiment.
>
> (4) Novel insights about brain function: We acknowledge the lack of novel insights about brain functioning in our work; however, our results that suggest the improved robustness and boundary detection performance of a full V1Net (relative to various lesions of the horizontal connections and their nonlinearity) can be observed as a re-validation of the importance of horizontal connections to computer vision models (in addition to biological vision).
>
> (5) Clarity issues on the paper's objectives and contributions: Our apologies for the lack of clarity in this area of our paper. Our contribution was to test whether horizontal connections play a role in improving the perceptual robustness and early visual task performance of artificial visual representations. To test this hypothesis, we proposed V1Net, a novel model of horizontal connections that is inspired by previously proposed small-scale models (which don’t scale well to today’s computer vision benchmarks). Being a model that can be easily incorporated into currently existing DCN implementations, we hope to encourage the computer vision community to utilize this simple addition to existing DCNs for improving robustness and early visual task performance.
>
> References:
> 1. Thomson, M. G., Foster, D. H., & Summers, R. J. (2000). Human sensitivity to phase perturbations in natural images: a statistical framework. Perception, 29(9), 1057-1069.
> 2. Tadmor, Y., & Tolhurst, D. J. (1993). Both the phase and the amplitude spectrum may determine the appearance of natural images. Vision research, 33(1), 141-145.

---

### Official Review · AnonReviewer3 · 2019-10-23
**Official Blind Review #3**

**Rating:** 1

**Review:**

The paper proposes, inspired by the Primates brain, to add horizontal inhibitory and excitatory connections. In practice, the work proposes a variant of convolutional LSTM cells, that incorporates additional convolutions.

Overall, the paper is hard to read, the experimental setting does not fully convince.  It seems hard to reproduce the results using the information given in the paper. A lot of the method section focuses on intuition, using vague vocabulary which makes it hard to understand concretely what is done in practice. In particular, the contributions are not clear enough. The difference between the proposed approach and existing works needs to be made clearer.
The idea is interesting, however, and the paper would benefit greatly from addressing those issues.



Main points


The claims seem somewhat bigger than the actual contributions. In practice, the contribution is a modification to an LSTM cell.

The paper is not easy to read, and mixes various terms without introducing them (e.g presynaptic activity is used to introduce the method but never introduced, not even in related work). It would be good to use standard notation and math font (e.g. small bold letter for vectors, etc) and to define notations. In Figure 1, the kronecker symbold represents convolutions but in (1), the hadamard product (*) represents convolutions. These inconsistensies make the paper harder to follow.

The problem of clarity extends to the experimental section, where the experiments are not clearly explained,

The method section could be more detailed, in particular, a mathematical comparison of LSTM vs the proposed approachh would be useful. Figure 1 is unclear: what do the red, dark blue and light blue lines correspond to mathematically?

The related attempts in ML and Deep learning should be reviewed ( e.g. Lateral Inhibition-Inspired Convolutional Neural Networkfor Visual Attention and Saliency Detection, AAAI 2018).

The experimental setting is not convincing. It seems that a simple state-of-the-art CNN architecture would do better than the proposed approaches. Comparing a ConvLSTM with a single convolution does not seem fair.

Very little implementation details are given. The authors mention a ConvLSTM2D layer is used. However, the inputs are static images: how exactly was the experiment done? Where does the time come from since the dataset considered is static?

About emergent neurally plausible horizontal connections: that section is interesting but would benefit from being more detailed and rigorous. There is no actual measure or study of the emergence of said connections.

3.2 is misleading as it corresponds to a setting that is never used in the experimental setting.

Most of the results are qualitative, except for Table 1. It would be useful to have quantitative comparisons on established benchmarks.

**Experience Assessment:**

I have read many papers in this area.

**Review Assessment: Checking Correctness Of Derivations And Theory:**

I assessed the sensibility of the derivations and theory.

**Review Assessment: Checking Correctness Of Experiments:**

I carefully checked the experiments.

**Review Assessment: Thoroughness In Paper Reading:**

I read the paper at least twice and used my best judgement in assessing the paper.

---

> ### Author Response · Authors · 2019-11-15
> **Official authors response to R3**
>
> We thank you for taking the time in carefully reviewing our submission. We shall address your concerns with regard to our work below:
>
> (1) Reading clarity issues: Thanks for pointing out the clarity issues in our paper, we elaborate more on terms such as the one mentioned in your review in order to ease the paper reading. We have also made the symbol usage consistent in our uploaded revision's figures and equations.
>
> (2) Comparison to standard ConvLSTM: This is a nice point, thank you. We have added the standard ConvLSTM equations next to V1Net’s equations.
>
> (3) Clarification on colored lines in Fig. 1: Following the convention used in neuroscience literature [3], our red, dark and light blue lines correspond to linear excitation (W_exc), nonlinear shunting (W_shunt) and linear subtractive inhibition (W_inh) operations respectively. We have made this clear by explicitly mentioning the weight variable names in our figure.
>
> (4) Reg. related work in ML/AI on lateral connections: Thanks for pointing this out, we have added the mentioned and additional related work from the ML/AI community on learning spatial dependencies.
>
> (5) Implementation details + image input to RNNs: Our apologies for the missing details. Similar to [2], we use activations from the bottom layer to set X_t for all timesteps. We would like to note that while the input is static, dynamics in response are possible due to the horizontal recurrent connections that unroll in time. We have added this information and more elaborate implementation details to our revision (in Appendix A.1).
>
> (6) About emergent neurally plausible horizontal connections: Thanks for your interest in this line of our ongoing work. To add more detail to Fig. 6, we notice strong (sometimes oriented) excitation among kernels that detect similar features. This observation is validated by previous experimental neuroscience findings that report a similar like-to-like excitation among simple cells with similar orientation tuning in the primate visual cortex. We would also like to point out the learning of other neurally plausible connectivity structures such as center-on surround-off and border-ownership [1] cells in our V1Net model.
>
> (7) Quantitative analyses: We have additionally compared our model's boundary detection performance to the current state of the art boundary prediction methods along with their model parameter count. We hope you value the comparable performance of our model (w/ 500k parameters) to other SOTA models with orders of magnitude more number of parameters.
>
> References:
> 1. Hesse, J. K., & Tsao, D. Y. (2016). Consistency of border-ownership cells across artificial stimuli, natural stimuli, and stimuli with ambiguous contours. Journal of Neuroscience, 36(44), 11338-11349.
>
> 2. Zamir, A. R., Wu, T. L., Sun, L., Shen, W. B., Shi, B. E., Malik, J., & Savarese, S. (2017). Feedback networks. In Proceedings of the IEEE Conference on Computer Vision and Pattern Recognition (pp. 1308-1317).
>
> 3. Pastore, V. P., Massobrio, P., Godjoski, A., & Martinoia, S. (2018). Identification of excitatory-inhibitory links and network topology in large-scale neuronal assemblies from multi-electrode recordings. PLoS computational biology, 14(8), e1006381.

---

### Official Review · AnonReviewer1 · 2019-10-30
**Official Blind Review #1**

**Rating:** 3

**Review:**

I first want to thank the authors for their proposed approach in this paper. Authors made an attempt to bridge the gap between natural (primate) vision and NN models. The paper is easy to read and understand. The authors proposed a Conv-LSTM-inspired model called V1Net. The model shows some merits in detecting the correct labels for noised inputs.

Unfortunately, the paper lacks some critical analysis, and V1Net usability is limited in real-life. Commonly, the community expects a certain number of experiments to back a claim. Specifically I have the following questions:

1. “V1Net can be flexibly incorporated as a module in existing implementations of DCNs”, can you elaborate how? Current architecture rarely consider using conv-lstm to solve tasks such as object detection. Not saying the current trend is correct or incorrect in doing so, but lack of experiments and details leave this claim unsupported.
2. The leap between horizontal connectivity and V1Net is rather unmotivated. Specifically, authors should explain why V1Net is the only (or the most suitable) way of implementing horizontal connectivity.
3. Why is the FFT texturization the correct way to evaluate the robustness? Specifically, could it be that V1Net being a simpler model (such as the case in Fig 4, where the performance is lower than conv-lstm for clean data) simply generalizes better to noisy input. Simpler models sometimes have the tendency to remove noise better.
4. Necessary comparisons are not made with at least a few state of the art models in CIFAR. As it stands the impact is limited on community who uses conv-lstm for CIFAR classification.
5. Figure 4 lacks the required standards of a scientific article figure. Borders on only 3 sides, and DPI seems to be low as text is pixelized.

**Experience Assessment:**

I have published in this field for several years.

**Review Assessment: Checking Correctness Of Derivations And Theory:**

I carefully checked the derivations and theory.

**Review Assessment: Checking Correctness Of Experiments:**

I carefully checked the experiments.

**Review Assessment: Thoroughness In Paper Reading:**

I read the paper at least twice and used my best judgement in assessing the paper.

---

> ### Author Response · Authors · 2019-11-15
> **Official authors response to R1**
>
> We thank you for taking the time in carefully reviewing our submission. We shall address your concerns with regard to our work below:
>
> (1) By stating ‘incorporation of V1Net within existing DCN implementations’, we mean to use V1Net as an additional layer (similar to Batch Normalization) that computes a nonlinear recurrent function of the input to each layer in the (deep) neural network. We conceptualize this as similar to normalizing the feature maps of a neural network’s layer L by taking into consideration a learned dynamic interaction between the different feature channels in that layer.
>
> (2) We have developed V1Net as a modification to a ConvLSTM; this is due to the success of a relevant prior model [3] that learns recurrent functions of static images, also derived from ConvLSTMs. By incorporating linear and nonlinear contextual interactions in a ConvLSTM, we believe that V1Net introduces a relatively simple technique to mix the computations of long-range horizontal connections (found in visual neuroscience literature; such as surround inhibition and excitation) into DCNs without deviating largely from standard deep learning modules that have been performing well on recent ML benchmarks. We do not claim V1Net to be the most suitable formulation of horizontal connections, however our learned neurally plausible connection structure (shown in Fig. 6.) seems to show evidence for V1Net’s similarity to biological horizontal connections.
>
> (3) Thanks for raising the interesting concern of model simplicity, we did not attempt to make V1Net simpler than a traditional Convolutional LSTM in our experiments. We have compared V1Net to both parameter-matched and receptive-field matched baseline ConvLSTM models on the Texturized-Challenge and BSDS500, and the results suggest that V1Net may not be computationally more simple than ConvLSTM models. These comparisons resulted in ConvLSTMs and V1Net obtaining roughly the same accuracy on the clean version of Texturized challenge across multiple random initializations, and V1Net obtains a better performance on the BSDS500 benchmark.
>
>
> We proposed to use FFT texturization as one way to evaluate robustness as we find evidence in previous vision science literature on how phase information in images is crucial for humans to perceive natural images [1,2] relative to amplitude/magnitude information. We do not intend to claim that FFT texturization is the objectively correct way to evaluate perceptual robustness, however our choice of such a manipulation is explained by previous use of this manipulation in vision science literature and the robustness of humans to this manipulation.
>
> (4) We acknowledge the lack of comparison to state-of-the-art models on CIFAR10, we are working on this and will add this as part of the revision in the future.
>
> (5) Thanks for pointing out the issue with our CIFAR results figure, we have updated this figure in our revision.
>
> References:
> 1. Thomson, M. G., Foster, D. H., & Summers, R. J. (2000). Human sensitivity to phase perturbations in natural images: a statistical framework. Perception, 29(9), 1057-1069.
> 2. Tadmor, Y., & Tolhurst, D. J. (1993). Both the phase and the amplitude spectrum may determine the appearance of natural images. Vision research, 33(1), 141-145.
> 3. Zamir, A. R., Wu, T. L., Sun, L., Shen, W. B., Shi, B. E., Malik, J., & Savarese, S. (2017). Feedback networks. In Proceedings of the IEEE Conference on Computer Vision and Pattern Recognition (pp. 1308-1317).

---

### Decision · Program_Chairs · 2019-12-19

**Decision:**

Reject

**Comment:**

The paper proposes a neurally inspired model that is a variant of conv-LSTM called V1net. The reviewers had trouble gleaning the main contributions of the work. Given that it is hard to obtain state of art results in neurally inspired architectures, the bar is much higher to demonstrate that there is value in pursuing these architectures. There are not enough convincing results in the paper to show this. I recommend rejection.